# Laboratory Tests on Swelling Properties of Field-Coring Gypsum Rock in Tunnels

Chongbang Xu [1], Haoju Fan [2], Xu Zhao [2,*], Lifeng Fan [2] and Peng Wang [2]

1   Bridge and Tunnel Research Center, Research Institute of Highway Ministry Transport, Beijing 100088, China
2   College of Architecture and Civil Engineering, Beijing University of Technology, Beijing 100124, China
*   Correspondence: zhaoxu@bjut.edu.cn

**Abstract:** The reaction between gypsum rock and water results in swelling deformation and swelling pressure. Swelling deformation and swelling pressure cause damage to underground engineering such as tunnels. It is of significance to study the swelling characteristics of gypsum rock. The variation in maximum radial free-swelling ratio, axial free-swelling ratio, lateral restricted-swelling ratio and lateral restricted-swelling pressure of gypsum rock with a water immersion time of 2880 min were investigated experimentally. The early swelling characteristics were further discussed and described by an S-shaped model. The results show that the swelling ratio and swelling pressure increase rapidly as the immersion time increases for the first 120 min. Subsequently, the swelling ratio and swelling pressure increase slowly and become stable as the immersion time further increases. At the 120th minute, the maximum radial free-swelling ratio, axial free-swelling ratio, lateral restricted-swelling ratio and lateral restricted-swelling pressure of gypsum rock reach more than 80% of their final values (2880 min in the present study). Based on the swelling characteristics of gypsum rock during the first 120 min, an S-shaped swelling-time model was introduced to describe the early swelling behavior of gypsum rock.

**Keywords:** gypsum rock; swelling behavior; laboratory test; swelling model

## 1. Introduction

Gypsum rock is deposited in many geological ages. The main components of the rock are anhydrite and gypsum. By the invasion of water, the anhydrite component ($CaSO_4$) can be hydrated in gypsum rock (the main component is $CaSO_4 \cdot 2H_2O$) [1,2]. In this procedure, swelling deformation and swelling pressure are developed in the rock mass. The swelling deformation and swelling pressure of gypsum rock may cause damage, which finally affects the stability of the engineering site [3–6]. Therefore, it is of practical significance to study the swelling characteristics of gypsum rock [4,5,7,8].

The swelling of gypsum rock is the result of coupling effects of physical swelling and chemical swelling [9]. Because of the physical and chemical coupling, the process of complete hydration of anhydrite into gypsum lasts a long time and produces swelling deformation and swelling pressure [10–12]. By studying the hydration and swelling mechanism of anhydrite, a 62.6% increase in molar volume was found during the hydration of anhydrite into gypsum [13,14]. The maximum swelling pressure of the sample in the Gipskeuper formation was found to reach 16.0 MPa [15]. In order to study the growth of swelling pressure during the hydration of anhydrite into gypsum, a swelling pressure test was carried out for more than two years. Swelling pressure increased rapidly in the early stage of the test, and in the first few hours of the experiment, several large-scale increases were observed in the test [15–17]. Some scholars believed that the swelling pressure at the engineering site may be smaller than the value measured in the laboratory because of the certain scale effect [7,15,18]. In order to measure the swelling pressure at the engineering construction site, pressure cells were installed inside the Lilla tunnel to record changes

in swelling pressure over time. During the monitoring period of more than one year, the swelling pressure did not reach 16.0 MPa, but the overall swelling law was similar to laboratory research. The swelling pressure increased rapidly in the early stage, and then gradually became slower [4]. In the long-term test of swelling characteristics in these laboratories and fields, the swelling pressure was different, but its overall swelling law was similar. A rapid growth of the swelling force in the early stage of the test was observed.

In some tunnel engineering, due to groundwater or construction water providing enough water for gypsum-rock hydration reaction, the tunnel could be destroyed by the swelling of gypsum rock in a short time. For example, the Chienberg road tunnel suffered severe swelling and damage in a short period of time during the excavation [11]. Some scholars have studied the swelling characteristics of mudstone, shale and other rocks. Some strength tests were carried out on the swelling mudstone samples with different water content, and a complete empirical formula for predicting the mechanical properties of swelling mudstone was established [19]. Wetting tests were conducted using an environmental scanning electron microscope (ESEM), which addresses the swelling property of argillaceous rock under resaturated conditions [20]. A solid model with cohesive elements based on the finite element method was used to predict the generation of micro-cracks in shale rock [21].

The swelling of the existing research on gypsum rock characteristics is of great significance. However, these studies have focused on the long-term expansion properties of gypsum rock and are very time-consuming, both for laboratory experiments and field tests. Additionally, there are few studies on the swelling characteristics of gypsum rock in the short term. In this study, an S-shaped swelling-time model was put forward through laboratory experiments to explore the early swelling characteristics of gypsum rocks. Studying the short-term swelling characteristics of gypsum rock can help to better understand the swelling characteristics of gypsum rock and can provide some reference for preventing early swelling disasters.

## 2. Preparation

### 2.1. Sample Preparation

In this study, the original rock of the sample is collected from a tunnel jobsite in Lvliang City, Shanxi Province, China. The tunnel structure has suffered a rock swelling disaster. The destruction of the tunnel is shown in Figure 1. According to the X-ray diffraction analysis, the main mineral composition of the gypsum rock is 75% gypsum, 19.8% anhydrite and 5.2% quartz, as shown in Figure 2.

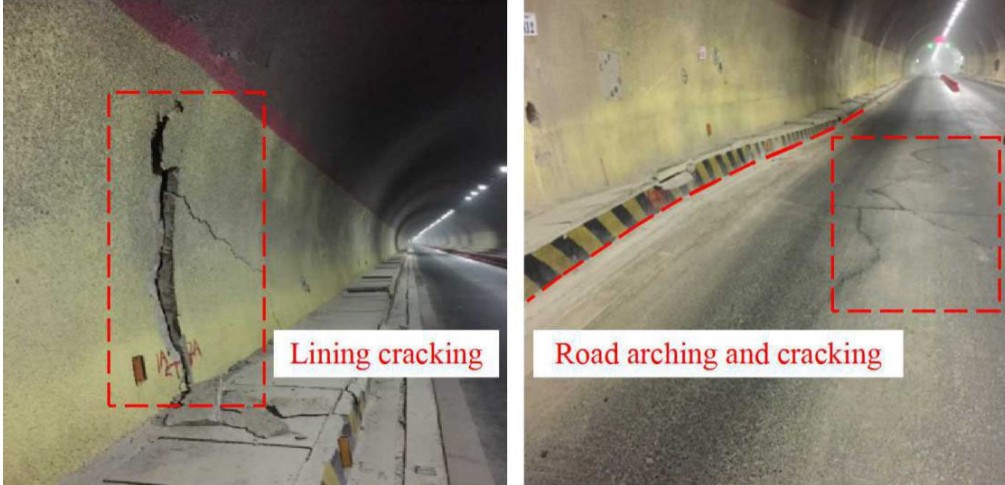

**Figure 1.** Swelling disaster of Dugongling tunnel in Shanxi province.

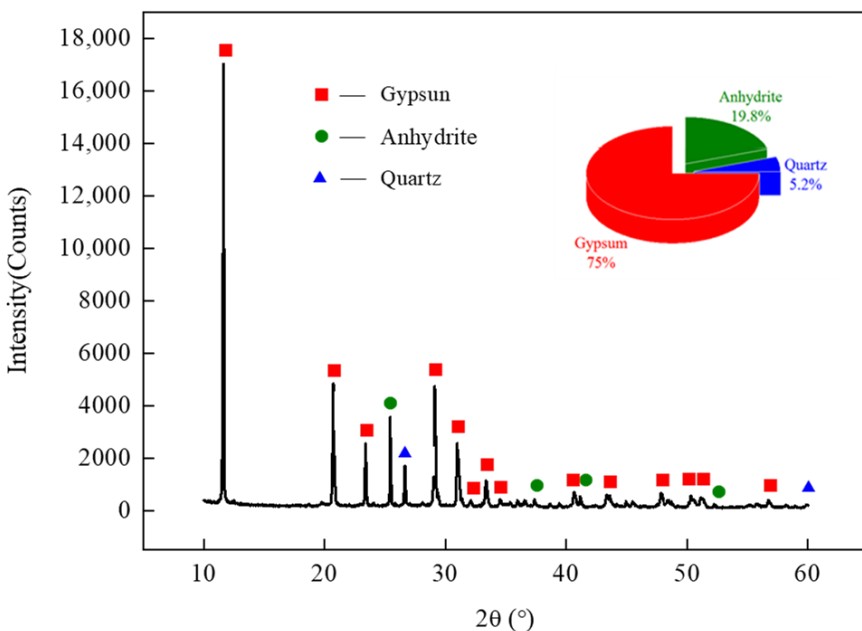

**Figure 2.** Mineral components of present gypsum rock.

Rock core-cutting technology was adopted to process the samples, and dry-wind cooling was used to avoid pre-swelling of the sample during processing. The gypsum samples to be tested were drilled from an original rock in parallel directions to avoid the influence of rock structures of different layers.

Three groups of samples are used in this study, as shown in Figure 3, for the free-swelling ratio test, the lateral restricted-swelling ratio test and the lateral restricted-swelling pressure test. The left column in Figure 3 shows group 1, with a diameter of 50.0 mm and a height of 50.0 mm, for the free-swelling ratio test, numbered PA-1, PA-2 and PA-3, respectively. The middle column in Figure 3 shows group 2, with a diameter of 50.0 mm and a height of 25.0 mm, for the lateral restricted-swelling ratio test, numbered PB-1, PB-2 and PB-3, respectively. The right column in Figure 3 shows group 3, with a diameter of 50.0 mm and a height of 25.0 mm, for the lateral restricted-swelling pressure test, numbered PC-1, PC-2 and PC-3, respectively. The dimensions of the samples are shown in Table 1.

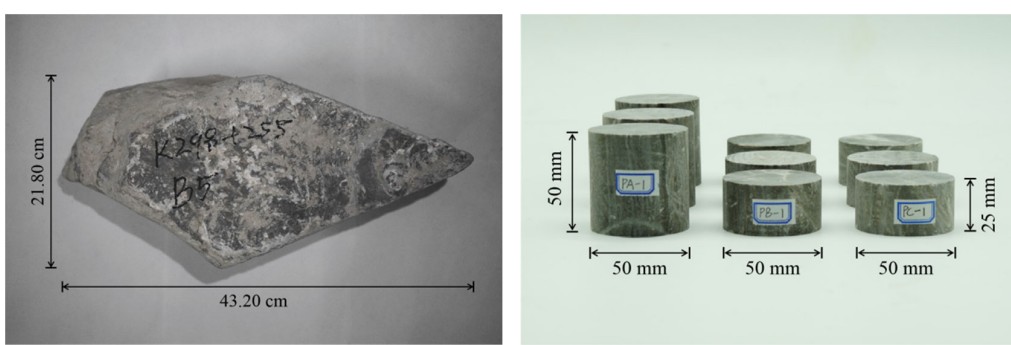

**Figure 3.** Gypsum rock and laboratory rock samples.

**Table 1.** Dimensions of gypsum rock samples.

|         | Size              | Number          | Classification of Test               |
|---------|-------------------|-----------------|--------------------------------------|
| Group 1 | Φ50 mm × 50 mm    | PA-1, PA-2, PA-3 | free-swelling ratio                  |
| Group 2 | Φ50 mm × 25 mm    | PB-1, PB-2, PB-3 | lateral restricted-swelling ratio    |
| Group 3 | Φ50 mm × 25 mm    | PC-1, PC-2, PC-3 | lateral restricted-swelling pressure |

### *2.2. Dehydration*

To remove free water and crystallization water from the gypsum rock samples, a MXQ1200-50 box heating furnace was used to progress high temperature dehydration. The samples were placed in the furnace and heated to 220 °C with a heating ratio of 5.0 °C/min, and were then kept at a constant temperature for 2880 min. Finally, the furnace was turned off and the samples were cooled to room temperature. After this procedure, the rock sample color changed to a light gray, the bedding was clearly visible, and the surface was smooth with no obvious cracks.

### 3. Experimental Investigation

### *3.1. Free-Swelling Ratio Test*

Rock free-swelling ratio is defined as the ratio of the radial (axial) deformation of the rock sample after immersion in water to the original radial (axial) dimension of the sample. The three samples of group 1 were tested. The YS-II rock-swelling ratio setup was produced as shown in Figure 4. It can be seen from Figure 4 that there were four dial indicators in the horizontal direction, numbered dial indicator 1, 2, 3 and 4, respectively. These dial indicators were used to record the swelling and deformation in the horizontal direction. There was a dial indicator in the vertical direction, numbered dial indicator 5. This dial indicator recorded the axial swelling ratio of the sample. The radial deformation and the axial deformation could be tested simultaneously by this device.

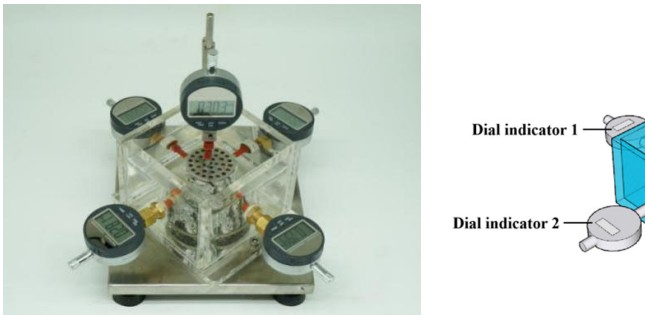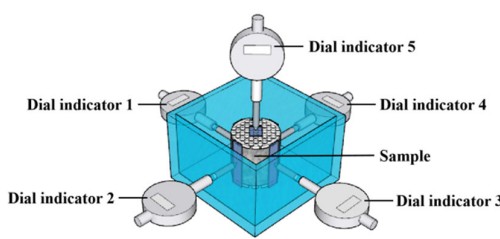

**Figure 4.** YS-II rock-swelling ratio setup.

During the test, 2 pieces of permeable stone were placed both at the upper and lower ends of the sample, respectively. To ensure the flatness of the contact surface between the top of the dial indicator and the sample, a metal block was placed on the axial direction. Dial indicators 1 and 3 were perpendicular to the direction of bedding; the recorded data were recorded as radial-swelling ratio 1. Dial indicators 2 and 4 were parallel to bedding direction; the recorded data were recorded as radial-swelling ratios 2 and 4, respectively. Dial indicator 5 recorded the axial-swelling ratio of the sample.

The dial indicators were calibrated before test. After the calibration, water was slowly poured into the container until the upper permeable stone was merged. Within the first hour, the dial indicator data were recorded every 10 min. During the testing period of the remaining 2820 min, the deformation and water immersion time were recorded every 60 min. The water temperature was kept with a variation within ±2.0 °C during the experiment. The water level was also kept stable.

The free-swelling ratio can be expressed as

$$V = \frac{\Delta}{D} \times 100\%, \tag{1}$$

where $V$ represents the free-swelling ratio (%), which can be axial free-swelling ratio and radial free-swelling ratio; $D$ represents the original dimension of the sample in the free swelling ratio test, which can be the original height or the original diameter of the sample;

and $\Delta$ represents the deformation of the sample in the free-swelling ratio test, which can be the axial deformation or the radial deformation of the sample.

Therefore, the axial free-swelling ratio can be expressed as

$$V_{H_F} = \frac{\Delta H_F}{H_F} \times 100\%, \tag{2}$$

and the radial free-swelling ratio calculation equation can be expressed as

$$V_{D_F} = \frac{\Delta D_F}{D_F} \times 100\%, \tag{3}$$

where $V_{H_F}$ represents the axial free-swelling ratio and $V_{D_F}$ represents the radial free-swelling ratio; $H_F$ represents the original height of the sample and $D_F$ represents the original diameter of the sample in the free-swelling ratio test; $\Delta H_F$ represents the axial-deformation value of the sample, and $\Delta D_F$ represents the radial-deformation value of the sample in the free-swelling ratio test.

### 3.2. Lateral Restricted-Swelling Ratio Test

Lateral restricted-swelling ratio refers to the ratio of the axial-swelling deformation of rock sample after immersion in water to the original axial dimension of the sample. The three samples of group 2 were tested in the YS-II rock-swelling ratio setup. The YS-II rock-swelling ratio setup was produced as shown in Figure 5.

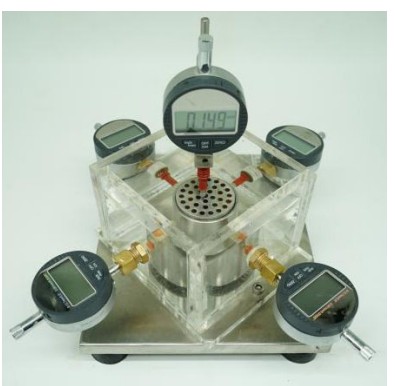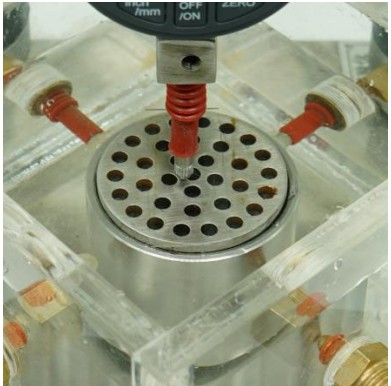

**Figure 5.** YS-II rock-swelling ratio setup with metal limit.

During the test, petroleum jelly was applied on the inner side of the round metal collar. Then, the sample was fixed into the metal collar for the constraint lateral deformation. Two pieces of permeable stone were placed both at the upper and lower ends of the sample, respectively.

After the dial indicators were calibrated, water was slowly poured into the container until the upper permeable stone was merged. Within the first hour, the dial indicator data were recorded every 10 min. During the testing period of the remaining 2820 min, the deformation and water-immersion time were recorded every 60 min. The water temperature was kept with a variation within 2.0 °C during the experiment. The water level was also kept stable.

The lateral restricted-swelling ratio can be expressed as

$$V_{H_L} = \frac{\Delta H_L}{H_L} \times 100\%, \tag{4}$$

where $V_{H_L}$ represents the lateral restricted-swelling ratio, $H_L$ represents the original height of the sample in the lateral restricted-swelling ratio test, and $\Delta H_L$ represents the axial deformation of the sample in the lateral restricted-swelling ratio test.

### 3.3. Lateral Restricted-Swelling Pressure Test

A series of rock lateral restricted-swelling tests were designed to investigate the axial swelling pressure of rock after immersion in water with lateral constraints. Three samples with diameters of 50.0 mm and heights of 25.0 mm numbered PC-1, PC-2 and PC-3 were tested on the YYP-40 rock swelling-ratio setup. The YYP-40 rock swelling-ratio setup was produced as shown in Figure 6.

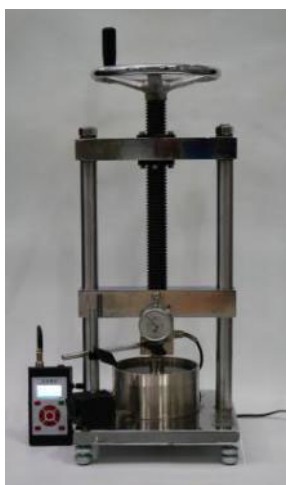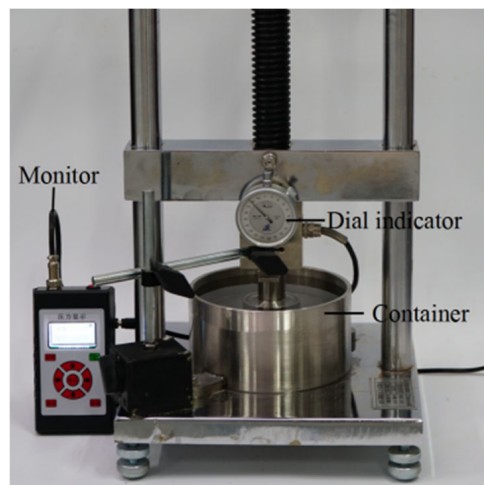

**Figure 6.** YYP-40 rock-swelling pressure setup.

During the test, petroleum jelly was applied on the inner side of the round metal collar. Then, the sample was fixed into the metal circle to constrain lateral deformation. Two pieces of permeable stone were placed both at the upper and lower ends of the sample, respectively. The dial indicator and the monitor were calibrated. The water was slowly poured into the container until the upper permeable stone was submerged. When the amount of deformation accumulated to more than 0.001 mm, pressure was loaded on the sample top to ensure a constant volume, thus the thickness of the test sample remained unchanged throughout the test. At the beginning, the test data were recorded every 10 min. When the difference between 2 adjacent data was smaller than 0.001 mm for 3 consecutive times, the deformation was considered to be stable and the test load was recorded. In order to further study the relationship between the swelling pressure and the immersion time, the swelling pressure was recorded at the same time points as in the swelling-ratio test. The immersion time was 2880 min. The water temperature change was smaller than $\pm 2.0\ ^\circ$C and the water level was kept constant during the experiment.

The lateral restricted-swelling pressure can be expressed as

$$p = F/A, \tag{5}$$

where $p$ represents lateral restricted-swelling pressure, $F$ represents the axial load and $A$ represents the cross-sectional area of the sample.

### 4. Results and Discussion

In the present study, gypsum rock samples after dehydration are used to study the swelling characteristics, the variation in free-swelling ratio, the lateral restricted-swelling ratio and the lateral restricted-swelling pressure of gypsum rock with a water immersion time within 2880 min were tested by using a rock-swelling-ratio setup and rock-swelling-pressure setup. The early swelling characteristics change laws are further discussed. Based on the swelling characteristics of gypsum rock during the first 120 min, an S-shaped swelling-time model was introduced to describe the early swelling behavior of gypsum rock.

### 4.1. Free-Swelling Ratio

The test results of free-swelling ratio are listed in Table 2 and plotted in Figure 7a–c, respectively. It can be seen from Figure 7a–c that the variations with time of the free-swelling ratios of the three samples are similar. It can be seen from Figure 7a that, during the first 120 min, the free-swelling ratios of the radial direction 1, the radial direction 2 and the axial direction of the samples PA-1, PA-2 and PA-3 all increase rapidly as the immersion time increases. Radial direction 1 represents the sum of the numerical changes in dial indicators 1 and 3, while radial direction 2 represents the other direction. The axial-direction swelling ratio is calculated by dial indicator 5, as shown in Figure 4. After the first 120 min, the free-swelling ratios of radial direction 1, radial direction 2 and the axial direction increase slowly as the immersion time increases. Similar to previous research, the largest value of the free-swelling ratios of radial directions 1 and 2 is defined as the maximum radial free-swelling ratio, which is used to show the free-swelling behavior parallel to the gypsum layer.

**Table 2.** Free-swelling ratio.

|  | Radial Direction 1 Swelling Ratio in the 2880th Minute (%) | Radial Direction 2 Swelling Ratio in the 2880th Minute (%) | Axial Direction Swelling Ratio in the 2880th Minute (%) |
|---|---|---|---|
| PA-1 | 10.86 | 4.53 | 6.02 |
| PA-2 | 14.50 | 4.94 | 5.71 |
| PA-3 | 12.10 | 6.82 | 7.03 |

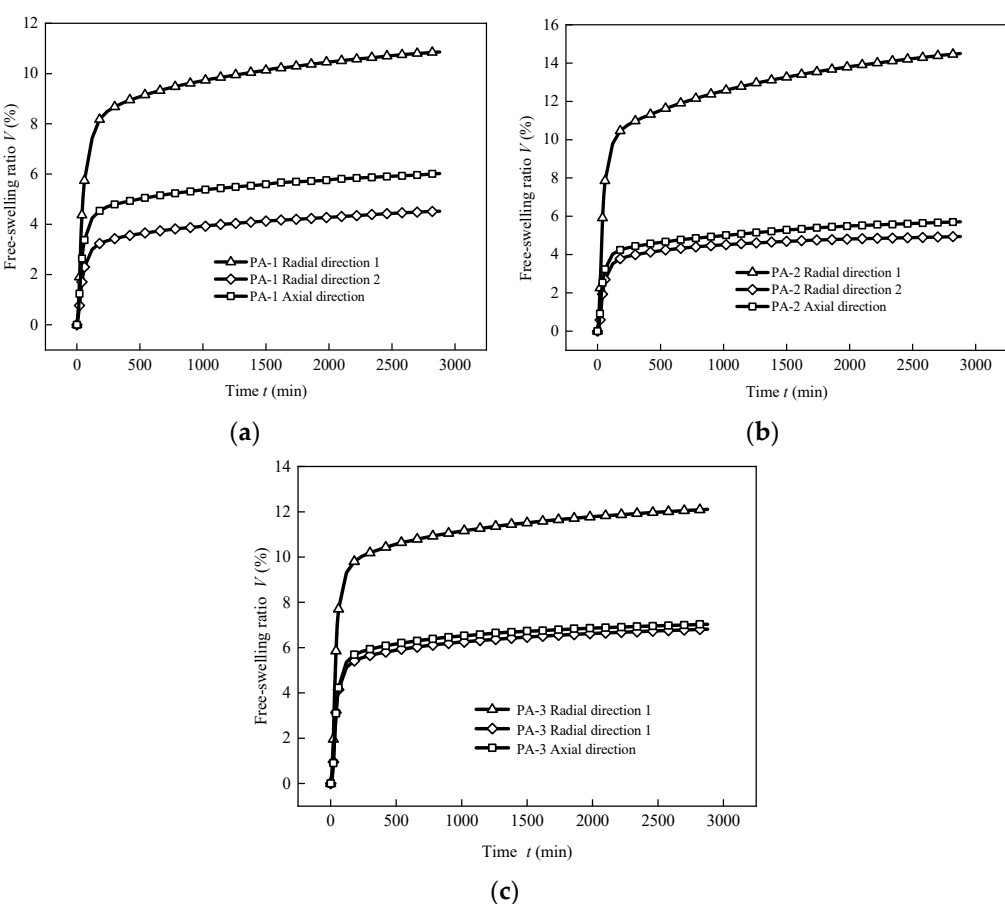

**Figure 7.** Free-swelling ratio: (**a**) Sample 1; (**b**) Sample 2; (**c**) Sample 3.

In order to further understand the maximum radial free-swelling deformation and axial free-swelling deformation of gypsum rock, the average value of the maximum radial

free-swelling ratio is achieved as shown in Figure 8. The average value of the axial free-swelling ratio is achieved as shown in Figure 9.

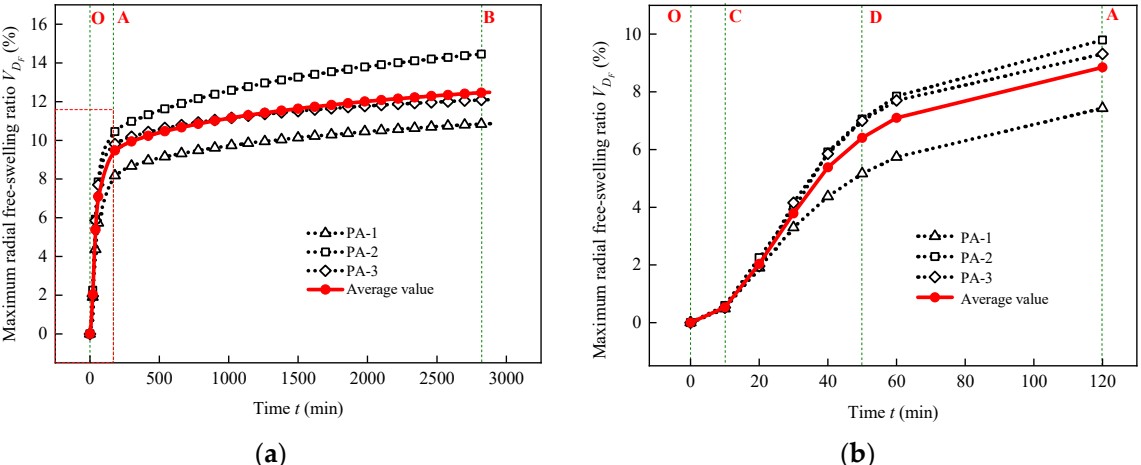

**Figure 8.** Maximum radial free-swelling ratio. Relationship between the maximum radial free-swelling ratio and time for (**a**) 2880 min and (**b**) during the first 120 min.

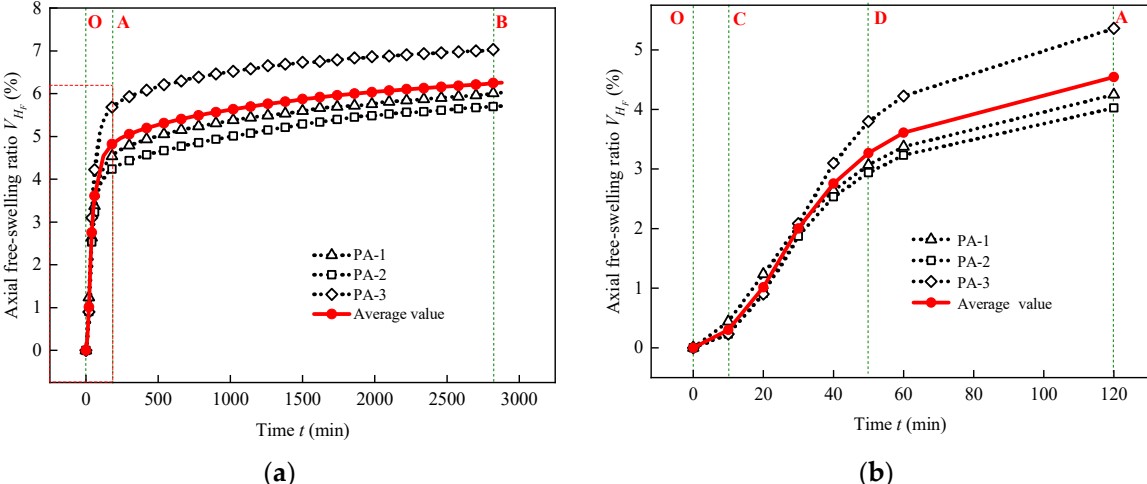

**Figure 9.** Axial free-swelling ratio. Variation in axial free-swelling ratio for (**a**) 2880 min and (**b**) during the first 120 min.

Figure 8 shows the maximum radial free-swelling ratio variation as time increases. It can be seen from Figure 8a that there is an inflection point when the maximum radial free-swelling ratio increases as time increases (t = 120 min in the present study). Before the inflection point, the maximum radial free-swelling ratio increases rapidly with the increase in immersion time. After the inflection point, the maximum radial free-swelling ratio becomes stable as immersion time further increases. The maximum radial free-swelling ratio at the inflection point is about 70.0% of the maximum radial free-swelling ratio at 2880 min.

Figure 8b shows the relationship between the maximum radial free-swelling ratio and the time for the first 120 min. It can be seen from Figure 8b that, as immersion time increases, the radial free-swelling ratio increases slowly at first, then rapidly, and then becomes slow again in the final period. The radial free-swelling ratio shows as an S-shaped increase trend with the increase in immersion time. The maximum radial free-swelling ratio reached 8.83% during the first 120 min.

Figure 9a shows the variation in axial free-swelling ratio with time. During the first 120 min, the axial free-swelling ratio increases rapidly with the increase in immersion time.

After 120 min, the axial free-swelling ratio increases slowly with the increase in immersion time. The maximum axial free-swelling ratio after immersion for 2880 min reaches 6.26%.

Figure 9b shows the variation in axial free-swelling ratio for the first 120 min. The axial free-swelling ratio also shows an S-shaped increasing tendency. The axial free-swelling ratio increases slowly at first, then rapidly, and slowly again later on. The axial free-swelling ratio reaches 4.54% at the 120th minute. Therefore, more than 73.0% swelling deformation had taken place after the first 120 min.

It can be seen from Table 3 and Figures 8a and 9a that the increase in the swelling ratio with time can be divided into two stages. The first stage is the first 120 min of the OA segment. During this period, the swelling ratio grows rapidly. The second stage takes place between the 120th and the 2880th minute of the AB segment. During this period, the swelling ratio grows slowly.

**Table 3.** Maximum radial free-swelling ratio.

| | Maximum Radial in the 2880th Minute (%) | Axial in the 2880th Minute (%) | Average Value of Maximum Radial in the 2880th Minute (%) | Average Value of Axial in the 2880th Minute (%) | Average Value of Maximum Radial in the 120th Minute (%) | Average Value of Axial in the 120th Minute (%) |
|---|---|---|---|---|---|---|
| PA-1 | 10.86 | 6.02 | | | | |
| PA-2 | 14.50 | 5.71 | 12.48 | 6.26 | 8.82 | 4.53 |
| PA-3 | 12.10 | 7.03 | | | | |

The early expansion of gypsum rock is carried out along all directions. However, the maximum radial free-swelling ratio occurs in the radial direction, while there is a significant difference between the axial free-swelling ratio and the maximum radial free-swelling ratio, which indicates that the swelling characteristics of gypsum rock are different in different joint directions. After the gypsum rock comes into contact with water, the rock surface expands at first, causing difficulty for the water to enter and slowing down the subsequent expansion behavior.

### 4.2. Lateral Restricted-Swelling Ratio

Table 4 shows the experimental results of the lateral restricted-swelling ratio test, which were plotted in Figure 10. It can be seen from Figure 10a that, in the initial stage of water immersion, the sample swells and the axial lateral restricted-swelling ratio increases rapidly with the increase in immersion time for the first 120 min. After 120 min, the axial lateral restricted-swelling ratio increases slowly as the immersion time further increases, reaching 3.07% after immersion for 2880 min.

**Table 4.** Lateral restricted-swelling ratio.

| | 2880th Minute (%) | Average Value in the 2880th Minute (%) | 120th Minute (%) | Average Value in the 120th Minute (%) |
|---|---|---|---|---|
| PB-1 | 3.64 | | 2.78 | |
| PB-2 | 3.02 | 3.36 | 2.47 | 2.58 |
| PB-3 | 3.42 | | 2.49 | |

Figure 10b shows the variation in axial lateral restricted-swelling ratio during the first 120 min. The axial lateral restricted-swelling ratio shows an S-shaped increase with the increase in immersion time. As immersion time increases, the swelling ratio increases slowly at first, then increases rapidly, and finally becomes constant. The radial-swelling ratio reaches 2.58% at the 120th minute, and about 76.8% of that after 2880 min.

It can be seen from Table 4 and Figure 10 that the increasing axial lateral restricted-swelling ratio with time can also be divided into two stages. The first stage is the first 120 min of the OA segment. During this period, the swelling ratio grows rapidly. The



second stage takes place between the 120th and the 2880th minute of the AB segment. During this period, the swelling ratio grows rapidly.

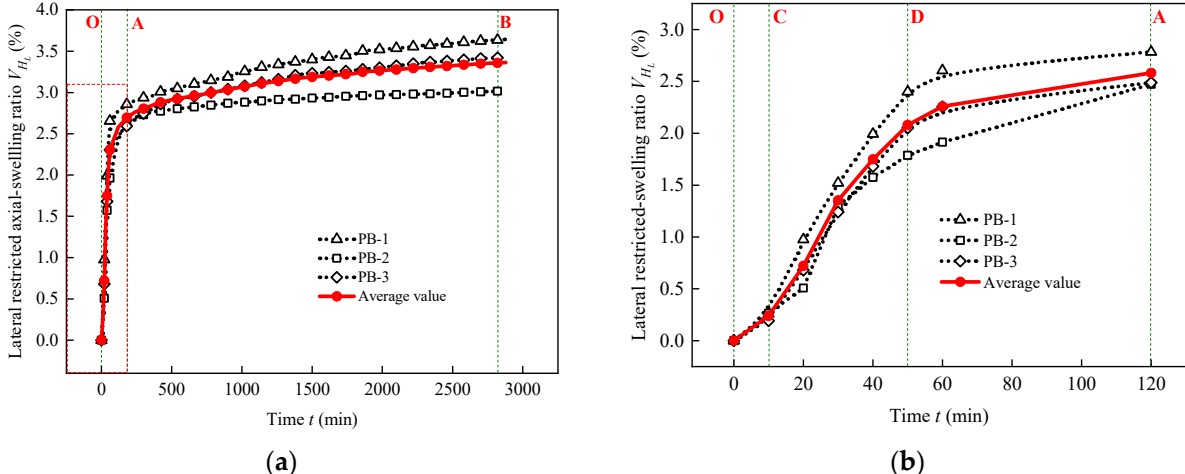

**Figure 10.** Axial lateral restricted-swelling ratio. Variation in axial lateral restricted-swelling ratio for (**a**) 2880 min and (**b**) during the first 120 min.

Compared with the free-swelling deformation, the swelling deformation of gypsum rock under lateral restricted conditions is smaller. It may be that the lateral restricted environment brings resistance to the expansion of gypsum rock and slows down the reaction between water and gypsum rock.

### 4.3. Lateral Restricted-Swelling Pressure

Table 5 shows the experimental results of lateral restricted-swelling pressure, which were plotted in Figure 11. It can be seen from Figure 11 that, in the initial stage of water immersion, the lateral restricted-swelling pressure increases rapidly. After 120 min, the lateral restricted-swelling pressure increases slowly with the increase in immersion time.

**Table 5.** Lateral restricted-swelling pressure.

|  | 2880th Minute (MPa) | Average Value in the 2880th Minute (MPa) | 120th Minute (MPa) | Average Value in the 120th Minute (MPa) |
|---|---|---|---|---|
| PC-1 | 5.19 |  | 4.00 |  |
| PC-2 | 4.74 | 4.75 | 4.17 | 3.71 |
| PC-3 | 4.32 |  | 2.94 |  |

Figure 11a shows the variation in lateral restricted-swelling pressure with time. During the first 120 min, its increasing trend is similar to the free-swelling ratio test and the lateral restricted-swelling ratio test. Lateral restricted-swelling pressure increases rapidly with the increase in immersion time. However, after 120 min, the lateral restricted-swelling pressure tends to be flat, and no obvious increase can be observed. The lateral restricted-swelling pressure after immersion for 2880 min reaches 4.75 MPa.

Figure 11b shows the variation in lateral restricted-swelling pressure during the first 120 min. The lateral restricted-swelling pressure shows an S-shaped increase with the increase in immersion time. As immersion time increases, the lateral restricted-swelling pressure increases slowly at first, then increases rapidly, and it finally becomes stable. The swelling pressure reaches 3.71 MPa at the 120th minute, which is 78.1% of the lateral restricted-swelling pressure after immersion for 2880 min. Similar to previous sections, it can also be seen from Table 5 and Figure 11a that the increasing lateral restricted-swelling pressure with time can be divided into two stages. The first stage is the first 120 min of

OA segment. During this period, the lateral restricted-swelling pressure grows rapidly. The second stage takes place between the 120th and the 2880th minute of the AB segment. During this period, the lateral restricted-swelling pressure grows rapidly. Under lateral restricted conditions, the swelling pressure shows similar laws to the swelling deformation.

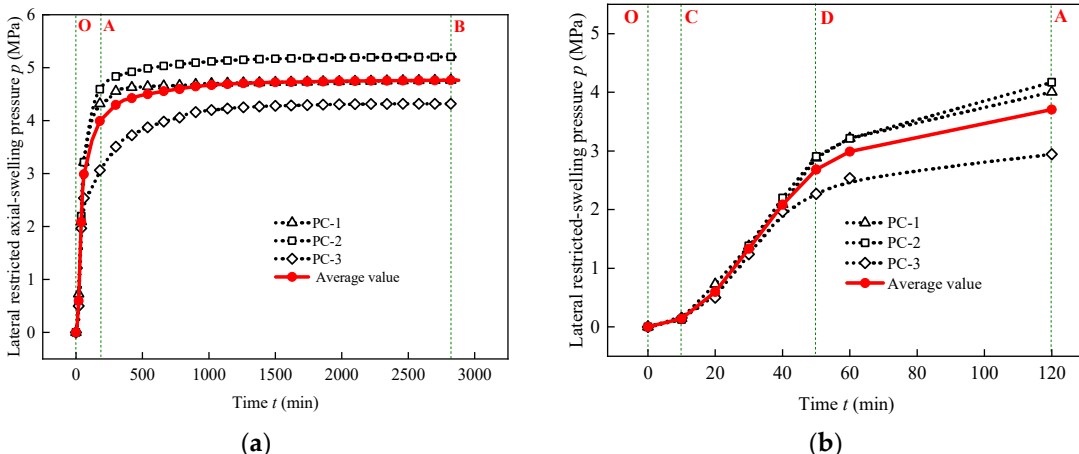

**Figure 11.** Lateral restricted-swelling pressure. Variation in axial lateral restricted-swelling pressure with time for (**a**) 2880 min and (**b**) during the first 120 min.

The early swelling deformation and swelling stress of gypsum rock change rapidly, and the expansion rate slows down with time. Compared with the free expansion, the expansion characteristics of gypsum rock with lateral restricted swelling are less expressed. The swelling stress occurs simultaneously with the swelling deformation, showing a similar law.

## 5. An S-Shaped Swelling-Time Model

Based on the previous analysis of the test results, the swelling ratio and swelling pressure with the immersion time within 2880 min has two main stages, which can be seen in Figures 8–11. The first stage takes place during the first 120 min of the OA segment. This stage is the rapid-growth stage. The second stage takes place between the 120th and the 2880th minute of the AB segment and it is a slow-growth and stable stage.

Around 120 min, at point A at the end of the first stage, the free-swelling ratio, the lateral restricted-swelling ratio and the lateral restricted-swelling pressure reach more than 70% of those after 2880 min. This explains the rapid growth of the OA segment. After expanding the OA segment, it can be seen from Figures 8b, 9b, 10b and 11b that the swelling ratio or swelling pressure has three main growth stages in the first 120 min. There are two obvious turning points in these three stages. They are point C, at about the 10th minute, and point D, at about the 50th minute. The first stage covers from minutes 0 to 10 of the OC segment. During this period, the swelling ratio grows slowly. During the second stage, from minutes 10 to 50 of the CD segment, the swelling ratio grows rapidly. The third stage, takes place from minutes 50 to 120 of the DA segment. During this period, the swelling ratio grows slowly and gradually flattens.

Gypsum rock often takes a long time to fully swell. However, in the early stage of immersion, the swelling ratio and swelling pressure increase rapidly. In view of the "swelling-time" characteristics and "S"-shaped growth characteristics of gypsum rock in the early stage of water immersion, a model was proposed to describe the early swelling behavior of gypsum rock [22]. The relationship between the swelling ratio and the swelling pressure with the immersion time can be expressed by

$$S(t) = A_2 - \frac{A_1}{1 + \left(\frac{t}{A_3}\right)^{A_4}}, \qquad (6)$$

where $t$ is the immersion time, $S(t)$ can be the maximum radial-swelling ratio, axial free-swelling ratio, lateral restricted-swelling ratio and lateral restricted-swelling pressure, and $A_1$, $A_2$, $A_3$ and $A_4$ are the parameters.

The S-shaped model was used to predict the swelling behavior of gypsum rock. Figure 12a–d were used to predict the maximum radial free-swelling ratio, axial free-swelling ratio, lateral restricted-swelling ratio and lateral restricted-swelling pressure, respectively. The fitting parameters are shown in detail in Table 6. It can be observed from Figure 12 and Table 6 that the S-shaped model can describe the swelling behavior of gypsum rock with a high accuracy. The correlation coefficients are as large as 0.99 in the present study.

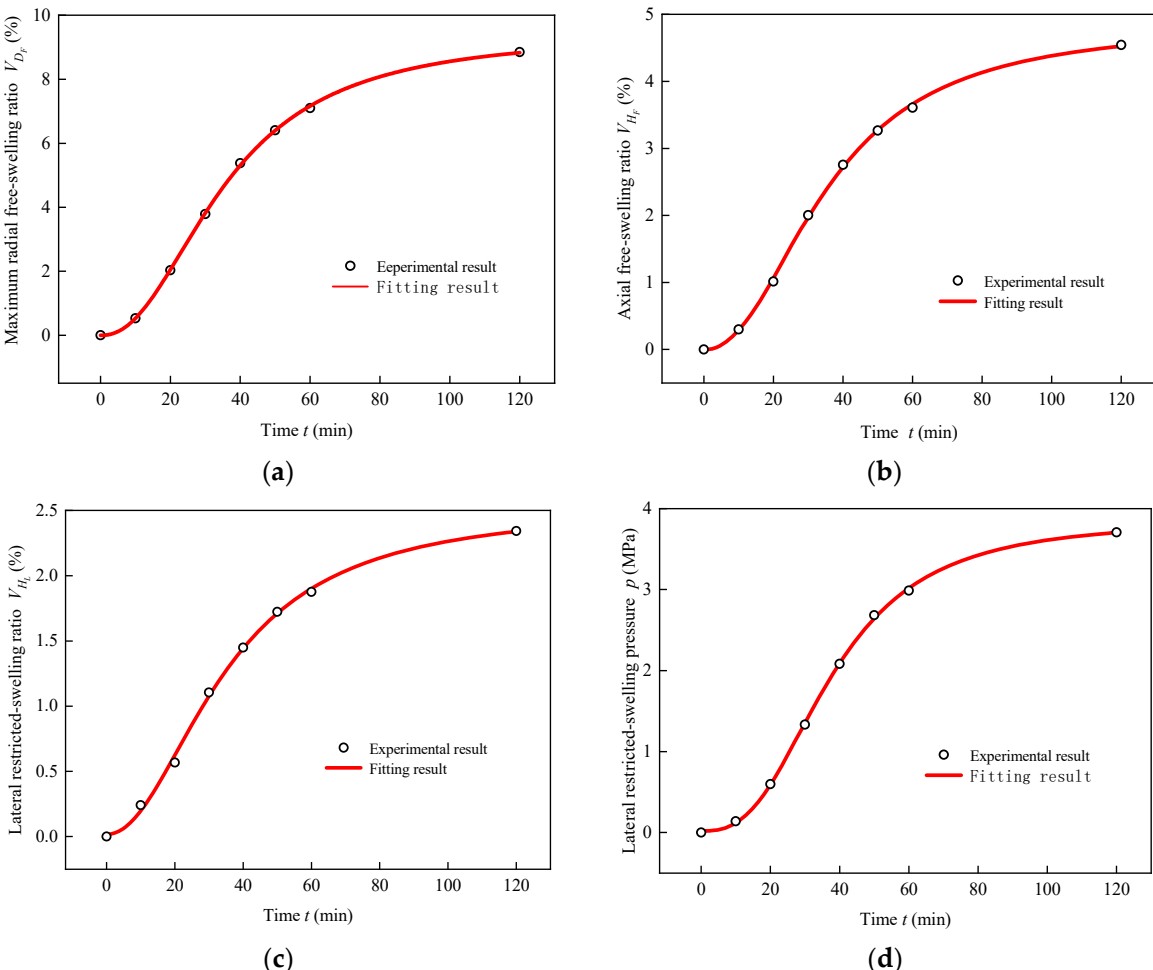

**Figure 12.** Swelling-time model. (**a**) Maximum radial free-swelling ratio; (**b**) axial free-swelling ratio; (**c**) lateral restricted-swelling ratio; (**d**) lateral restricted-swelling pressure.

**Table 6.** Parameters of S-model.

|  | $A_1$ | $A_2$ | $A_3$ | $A_4$ | $R^2$ |
|---|---|---|---|---|---|
| Maximum radial free-swelling ratio $V_{D_F}$ | 3.88 | 3.86 | 37.77 | 2.74 | 0.99 |
| Axial free-swelling ratio $V_{H_F}$ | 2.54 | 2.53 | 34.96 | 2.05 | 0.99 |
| Lateral restricted-swelling ratio $V_{H_L}$ | 9.42 | 9.43 | 35.69 | 2.12 | 0.99 |
| Lateral restricted-swelling pressure $p$ | 4.85 | 4.85 | 35.74 | 2.17 | 0.99 |

For gypsum rock composed of gypsum and other non-swelling minerals, the S-shape model can provide some reference for preventing early swelling disasters.

## 6. Conclusions

The early-age and long-term variation in maximum radial free-swelling ratio, axial free-swelling ratio and swelling pressure as immersion time increases were investigated, and an S-shaped model was introduced to describe the early-age swelling behavior of gypsum rock. The following conclusions were reached:

(1) The swelling ratio and swelling pressure of gypsum rock shows similar growth trends. In the early stage of immersion in gypsum rock, the swelling ratio and swelling pressure increased rapidly, and then gradually slowed down.

(2) The swelling characteristics of gypsum rock are different in different joint directions. Moreover, the lateral constraint environment will affect the expression of the swelling characteristics of gypsum rock.

(3) At the 120th minute of the test, the free-swelling ratio, the lateral restricted-swelling ratio and the lateral restricted-swelling pressure reach 72.3%, 76.8% and 78.1%, respectively, of those at the 2880th min. An S-shaped model can be used to describe the short-term swelling trend of gypsum rock with a high accuracy.

In this paper, the free-swelling ratio, the lateral restricted-swelling ratio and lateral restricted-swelling pressure of gypsum rock have been studied by means of laboratory tests. Some achievements have been made in this research, but the test conditions are relatively simple, and further research can be carried out from the joint of gypsum rock and dry–wet-cycle conditions in the future.

**Author Contributions:** Conceptualization, C.X., X.Z. and L.F.; data curation, P.W.; formal analysis, P.W.; investigation, H.F. and P.W.; visualization, P.W. and H.F.; resources, X.Z. and C.X.; writing—original draft, P.W.; writing—review and editing, H.F. and P.W. All authors have read and agreed to the published version of the manuscript.

**Funding:** National Transportation Pilot Project of Highway Research Institute of the Ministry of Transport (QG2021-1-3-3).

**Institutional Review Board Statement:** Not applicable.

**Informed Consent Statement:** Not applicable.

**Data Availability Statement:** Not applicable.

**Conflicts of Interest:** The authors declare no conflict of interest.

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
