# Peer review of "Laboratory Tests on Swelling Properties of Field-Coring Gypsum Rock in Tunnels"

_applsci, doi:10.3390/app13020719_

Round 1

Reviewer 1 Report

TITLE

The laboratory test implying the field tunnel should be identified in the title.

ABSTRACT

The abstract is well written.

INTRODUCTION

§ In this section, the review of the method background should be expanded.

§ The study's objective should be stated clearly.

METHODOLOGY

§ What are the reasons for having different sample sizes?

§ What is the difference between sample Group 2 and Group 3?

§ The purpose of sample classification (Group 1 - 3) must be clearly explained.

§ Normally, the temperature has an effect on rock swelling. What is the temperature that must be maintained during testing? Is this temperature of the field (the temperature of formation)?

§ The Radial direction 1, 2, and Axial direction used in Table 2 must be explained in the methodology section, and the sample schematic figure is supposed to be used for better understanding.

RESULTS AND DISCUSSION

§ This section's discussion should be carefully provided and improved; I couldn't find the results' discussion and interpretation in this section. The author only displayed the results of the tests.

§ How do the authors account for temperature effects in the field?

§ Why do the results have to have both free and restricted swelling?

§ The reliability of the S-shape swelling model applied to the other sites for gypsum rocks needs to discuss more. Should any field condition be considered?

§ The comparison among three group samples must be discussed and interpreted.

§ What is the gap in this study? What do the recommendations for further work?

CONCLUSION

The key findings from this research are supposed to highlight in this part. The outcomes are also provided in the conclusion section, which connects the study's objective.

Author Response

Dear Reviewer:

Thank you for your comments concerning our manuscript entitled “Swelling Deformation and Swelling Pressure of Field Coring Gypsum Rock in Tunnel” (ID: applsci-2020897). Those comments are all valuable and very helpful for revising and improving our paper, as well as the important guiding significance to our researches. Please refer to the attachment for specific reply.

We appreciate for your warm work earnestly, and hope that the correction will meet with approval. Once again, thank you very much for your comments and suggestions.

Yours sincerely,

Xu Zhao

Reviewer 2 Report

The authors investigated the swelling Deformation and pressure of gypsum rock in tunnel.  The authors performed a field study on gypsum rocks using cylindrical core samples. Based on this study, they developed a model to estimate the swelling behavior of the gypsum rocks, and its effect on the construction applications, particularly the tunnels.

This is important topic for geotechnical engineering and construction applications. The topic fits with the journal scope, but there are major revisions should be carried out before further considerations of the manuscript.

v The authors used three series; each of them includes three samples. In this case, no need to present the results of each sample, just show the average with error bars.

v The authors did not use a reference gypsum rock with more or less 100 gypsum. The reference is important to validate the theoretical model.

v Is the model valid only for gypsum portion of the rock or the other minerals? This theoretical model and this study in general is limited to gypsum rocks composed from gypsum and non-swelling minerals such as quarts …

1.     Keywords (i.e. Swelling pressure)

·       The title of the paper will be registered on search engines, so no need to duplicate any terminology already used in the paper’s title.

·       Table 1

·       Is there a justification for using these dimensions of the samples?

2.     Line 30

·       “Rock stability” instead of “ …stability of rock engineering”

3.     Section 4.1

·       If samples PA1, PA2, and PA3 supposed to represent one type of rocks, then you can use the average including the error bar.

4.     Figures 7, 8, and 9

·       These figures show that the maximum swelling capacity of the gypsum rocks due to water is achieved in the first 100 minutes.

5.     Table 5

·       Organize this table; the titles of the columns are so long.

·       References

·       Add recent references, only one reference from nineteen was published within 5 years.

Author Response

(The authors gave the same response as above.)

Reviewer 3 Report

The manuscript presents an experimental study on the early-stage swelling deformation and swelling pressure of gypsum rock, which has significant effects on tunnel instability. The experimental results are presented clearly. However, the study doesn't provide any novel information or technique but only reports the results of some swelling experiments. Considering two revision cycles have been completed, the reviewer recommends the paper as a potential publication but the manuscript type should be reconsidered to be a 'technical note' rather than an 'original article'.

Author Response

Dear Reviewer 3:

Thank you for your comments concerning our manuscript entitled “Swelling Deformation and Swelling Pressure of Field Coring Gypsum Rock in Tunnel” (ID: applsci-2020897). We are willing to take your suggestion to change the paper type to 'technical note'.

We appreciate for your warm work earnestly. Once again, thank you very much for your comments and suggestions.

Yours sincerely,

Xu Zhao

Round 2

Reviewer 1 Report

Introduction:

Lines 69–75 should be relocated to the results section; the goal, output, or outcome of this study should be stated there instead.

Methodology:

1. The size effect will not result in a different outcome, the authors did not explain. By choosing one size, the experimental size restriction can be overcome. Scientific evidence should be used to support the use of various sizes.

2. The authors' approach to the difference in temperature between laboratory and field circumstances is still not clear. Do the results reflect the field situation, and if so, why are they different from laboratory results in terms of temperature?

Results and Discussion:

The conversation in this part needs to be carefully written and enhanced. What real-world fields can the S-shape swelling model be used to? Are there any requirements for using the S-shape model? What may be inferred from the three group results? What findings have emerged from each and every result? What distinguishes this study from previous ones? Etc.

Author Response

(The authors gave the same response as above.)

Reviewer 2 Report

The authors have addressed all of my concerns

Author Response

Dear Reviewer:

Thank you for your comments concerning our manuscript entitled “Swelling Deformation and Swelling Pressure of Field Coring Gypsum Rock in Tunnel” (ID: applsci-2020897). 

Yours sincerely,

Xu Zhao